# Design and Analysis for Hypoid Gears with Ease-Off Flank Modification

**Qin Wang [1], Jinke Jiang [2],\*, Hua Chen [1], Junwei Tian [1], Yu Su [1] and Junde Guo [1,3],\***

[1] School of Mechatronic Engineering, Xi'an Technological University, Xi'an 710021, China; wq991514@163.com (Q.W.); hc156131@163.com (H.C.); junweitian@163.com (J.T.); suyu@xatu.edu.cn (Y.S.)
[2] Key Laboratory of Automotive Transportation Safety Techniques of Ministry of Transport, School of Automotive, Chang'an University, Xi'an 710064, China
[3] National United Engineering Laboratory for Advanced Bearing Tribology, Henan University of Science and Technology, Luoyang 471023, China
\* Correspondence: jiangjk@chd.edu.cn (J.J.); gjd0119@xatu.edu.cn (J.G.)

**Abstract:** An approach of ease-off flank modification for hypoid gears was proposed to improve the meshing performance of automobile drive axle. Firstly, a conjugate pinion matching with gear globally was developed based on gear meshing theory. Secondly, a modified pinion was represented by a sum of two vector functions determining the conjugate pinion and the normal ease-off deviations expressed by both predesigned transmission error function and tooth profile modification curves to change the initial contact clearance of the tooth. Thirdly, the best ease-off deviations were determined by optimizing the minimum amplitude of loaded transmission error (ALTE) based on tooth contact analysis (TCA) and loaded tooth contact analysis (LTCA). Finally, the results show that effective contact ratios ($\varepsilon_e$) are established by clearances both teeth space and of contact elliptical, and greatly affect ALTE. The $\varepsilon_e$ is a variable value with increasing loads for the tooth with modification. ALTE decreases with increasing $\varepsilon_e$. After $\varepsilon_e$ reaches the maximum, ALTE increases with increasing loads. The mismatch of the best ease-off tooth is minimal, which contributes to effective reduction in ALTE, thus significantly improving drive performance.

**Keywords:** hypoid gear; ease-off modified; ALTE; LTCA; effective contact ratio





## 1. Introduction

Hypoid gears are widely applied in power transmissions for intersecting and crossed axes, respectively. Stress concentration is likely to occur at the edge of the tooth due to assembly errors and deformations, which could affect the fatigue life of gears. As a result, the technology of tooth surface modification is adopted in processing. Tooth surface design and processing is always the key point of research for hypoid gears because of geometry-complicated tooth surface and difficult processing techniques. TCA [1] and LTCA [2] are applied to control the contact quality and dynamic performance for hypoid gear. Many works are concerned with tooth design and processing to eliminate edge-loading contact conditions and improve the contact performance of hypoid gears transmissions, such as local synthesis tooth [3], high-order transmission error tooth [4–6], real tooth [7,8], global synthesis tooth [9], hypoid gear transmission (HGT) cutting technique [10], etc. These methods are focused on correcting the tooth with parabolic transmission error based on a cradle-type machine, and can effectively reduce stress concentration at the edge of the tooth, but meanwhile could cause excessive mismatch of gear pairs. Therefore, the methods cannot fundamentally improve the dynamic meshing performances. Recently, in [11], a novel method of conjugated action for crowning in a face-hobbed spiral bevel and hypoid gear were driven through the spirac system to improve gear pair contact degree and meshing quality. Besides, in Ref. [12], a method of grinding a stick blade profile to cut hypoid gears was used to improve the machining accuracy of gear surface.

High-order tooth correction technology of spiral bevel gear [13–17] based on multi-axis computer numerical control (CNC) machine and machine-tool settings contribute to the processing tooth with complex modification, such as ease-off flank modification. Ease-off flank modification, which can directly demonstrate various mismatch relations between the modified tooth surfaces and conjugate tooth surface, was applied to tooth correction [18,19] for hypoid gear and spiral bevel gear. Ease-off flank modification approaches were developed according to given transmission error and contact area for spiral bevel gear in [20–22]. Besides, in [23–25], multi-objective tooth optimization was carried out for spiral bevel gears and hypoid gear with modification-based LTCA to improve the overall meshing quality. According to [26], an approach of ease-off flank modification with three-segment parabolic transmission error tooth are developed for bevel gear. In [27], an ease-off modification approach for hypoid pinions based on a modified error sensitivity was proposed to improve the meshing performances. In [28], a penetration-based gear contact model for accurate and numerically efficient TCA of spiral bevel gears was used to investigate the simulation of gear alignment errors. In [29], a procedure to determine the extreme relative curvatures between conjugate gear flanks was presented, and ease-off topography and unloaded transmission error were used to assess conjugate action between mating gear flanks in direct contact.

The above studies are focused on TCA. LTCA simulation was used for spiral bevel gear and hypoid gear with ease-off flank modification. In order to improve the meshing quality of the hypoid gear, this paper proposes a method of tooth design and aims to analyze the pinion tooth with free ease-off flank modification expressed by both predesigned transmission error function and tooth profile modification curves, and the influences of the actual coincidence degree on ALTE under loads is discussed.

## 2. Ease-Off Flank Modification Methodology for the Pinion

### 2.1. Tooth Representation of Pinion Globally Conjugated with the Gear

When the pinion tooth surface is fully conjugated with a gear tooth surface, the drive ratio is equal to the nominal drive ratio of the gear pair. The relationship between the meshing angle of gear $\theta_2$ and that of pinion $\theta_1$ is as follows:

$$\theta_2 = N_1/N_2(\theta_1 - \theta_{10}) + \theta_{20} \tag{1}$$

Here, $\theta_{10}$ is the meshing rotating angles of pinion and $\theta_{20}$ is the gear at the design reference points; while $N_1$ and $N_2$ is the teeth number of pinion and gear, respectively.

The coordinate system of gear and pinion meshing is shown in Figure 1, and detailed instructions of it are detailed in [20]. $S_s$, $S_r$, and $S_q$ are the reference coordinates, and $S_2$ is the gear coordinate system. Moreover, $V$ is the offset distance, while $H_1$ and $H_2$ are the axial setting. $\Sigma$ represents the shaft angle. Based on the space meshing theory and coordinate transformation, the gear tooth surface, which is considered as an imaginary cutter is transformed into the coordinate system $S_1$ of the pinion tooth surface, and then the position vector $r_{10}$ and normal vector $n_{10}$ of the pinion are determined by the following formula:

$$\begin{cases} r_{10}(u, \beta, \theta_1,) = M_{1p}(\theta_1)M_{pq}M_{qr}M_{rs}M_{s2}(\theta_2)r_2(u, \beta) \\ n_{10}(u, \beta, \theta_1,) = L_{1p}(\theta_1)L_{qp}L_{qr}L_{rs}L_{s2}(\theta_2)n_2(u, \beta) \\ f_1(u, \beta, \theta_1) = n_{10}\cdot(\partial r_{10}/\partial\theta_1) \end{cases} \tag{2}$$

Here, $r_2$ and $n_2$ are the position vector and normal vector of the gear, respectively; $r_{10}$ and $n_{10}$ are the position vector and normal vectors of pinion globally conjugated with the gear, respectively; and $r_{10}$ is taken with respect to $S_1$. $u$ and $\beta$ are parameters of the tooth surface; $M_{1p}$, $M_{pq}$, $M_{qr}$, $M_{rs}$, and $M_{s2}$ are coordinate transformation matrices; and $L_{1p}$, $L_{pq}$, $L_{qr}$, $L_{rs}$, and $L_{s2}$ are corresponding $3 \times 3$ submatrices. $f_1$ refers to the equation of meshing $f_1 = n_{10}\cdot v_{10}$, where $v_{10}$ is the relative velocity between the cutter blade and the work gear in coordinate system $S_1$.

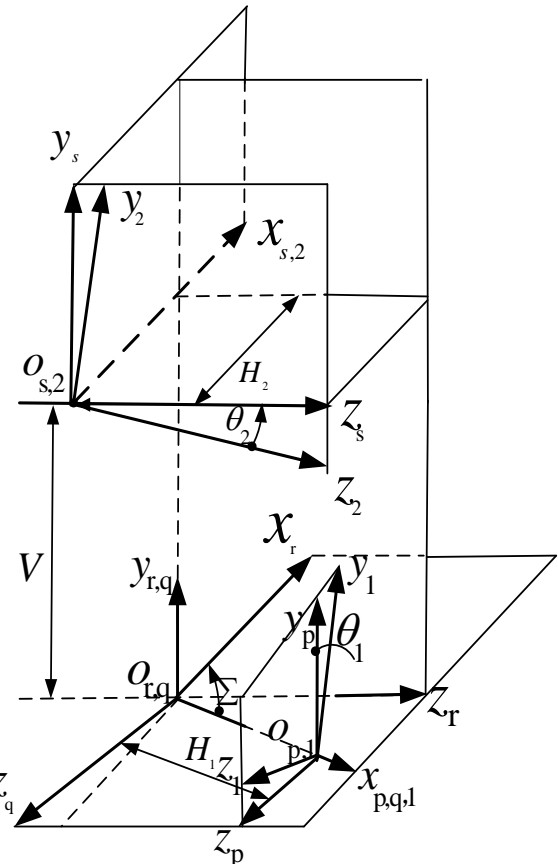

**Figure 1.** Generation of Hypoid Gear Coordinate System.

*2.2. Tooth Representation of Pinion with Ease-Off Flank Modification Only Depending on Transmission Error*

Tooth modification changes the initial contact clearance of conjugate tooth surface. The geometric transmission error reflects the initial clearance between contact pairs tooth. It does not change the length of contact line and contact path. Yet, the variation of the load distribution and loaded deformation between different meshing positions could pose a great impact on vibration (such as the profile modification of spur gears). The normal clearance of tooth surface can change the length of contact line and contact path to avoid certain edge stress concentration (for example, the longitudinal modification of spur gears), which are likely to affect the sensitivity of assembly error.

The modification design for tooth surface contact clearances follows the principles. ① In order to reduce the impact of meshing in and out, there should be enough parabolic transmission error near the meshing in and out end. ② In order to ensure that the meshing transformation point is as smooth as possible, and the loaded deformation at different meshing positions is basically the same, the backlash of several pairs of teeth in contact at the same time should not vary too much. In this way, there should be a certain transmission error in the middle and may form a concave shape (see Figure 2a), the expression is as follows:

$$\psi(\theta_1) = a_0 + a_1\theta_1 + a_2\theta_1^2 + a_3\theta_1^3 + a_4\theta_1^4 \qquad (3)$$

Here, Ψ is the geometric transmission error, $a_0$–$a_4$ can be solved by the data of point $p_0$–$p_4$ in Figure 2a, and $\varepsilon_1$–$\varepsilon_4$ is the undetermined parameter. According to the coincidence degree of the tooth surface, $\lambda$ and $t$ ($t > \lambda$, $0.5T \leq \lambda \leq T$, $T$ is the meshing period) can also be an undetermined parameter.

The expression of position vector $r_1$ and the normal vector of the pinion tooth surface $n_1$ only includes the predesign transmission error, which can be determined below:

$$\begin{cases} r_1(u, \beta, \theta_1,) = M_{1p}(\theta_1)M_{pq}M_{qr}M_{rs}M_{s2}(\theta_2)r_2(u, \beta) \\ n_1(u, \beta, \theta_1,) = L_{1p}(\theta_1)L_{qp}L_{qr}L_{rs}L_{s2}(\theta_2)n_2(u, \beta) \\ f_1(u, \beta, \theta_1) = n_1 \cdot (\partial r_1 / \partial \theta_1) \\ \theta_2 = N_1 / N_2(\theta_1 - \theta_{10}) + \theta_{20} + \psi(\theta_1) \end{cases} \tag{4}$$

The tooth surface normal clearance modification design adheres to the following principles. ① There needs to be certain tooth profile modification in the tooth root and tooth top to avoid edge stress concentration. ② The contact locus should also avoid the edge contact of the top of tooth top and its two sides, so there should be some distortion for the meshing in and out end (see Figure 2b). The modification curved surface can be expressed as $\delta_1(x_1, y_1)$. $x_1$ and $y_1$ are the radial $y_1$ and axial $x_1$ parameters of the pinion tooth surface. For the convenience of expression, the tooth profile modification curve in Figure 2c is obtained by rotation transformation mapping:

$$\delta_1 = \zeta(M_a(\theta_a))[0 \ y_1]^T \tag{5}$$

$$\begin{cases} \zeta(y_1) = e_0(y_1 - d_1)^4 & y_1 \le d_1 \\ \zeta(y_1) = e_1(y_1 - d_2)^4 & y_1 \ge d_2 \\ \zeta(y_1) = 0 & d_1 \ge y_1 \ge d_2 \end{cases} \tag{6}$$

Here, $d_1$, $d_2$, $q_1$, $q_2$, and $\theta_a$ are the undetermined modification parameters; the corresponding exponent could be two or four; and $M_a$ is the rotation transformation matrix.

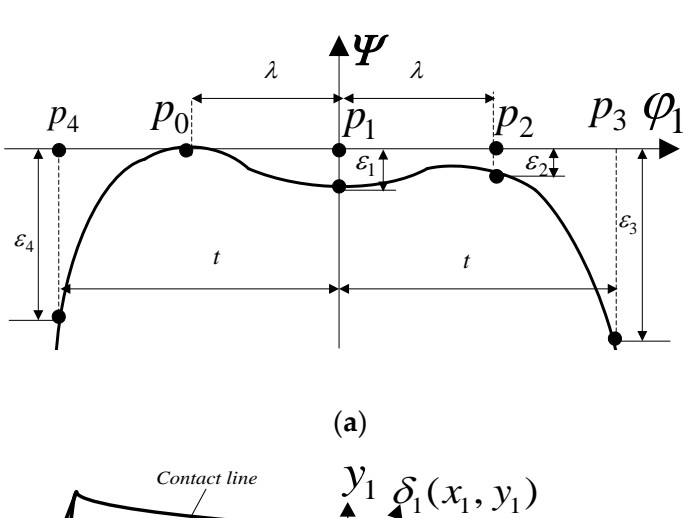

(a)

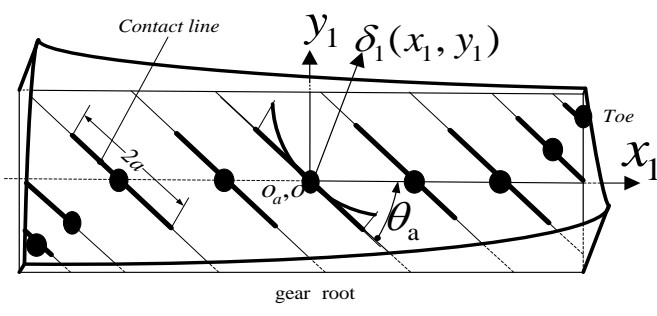

(b)

**Figure 2.** *Cont.*

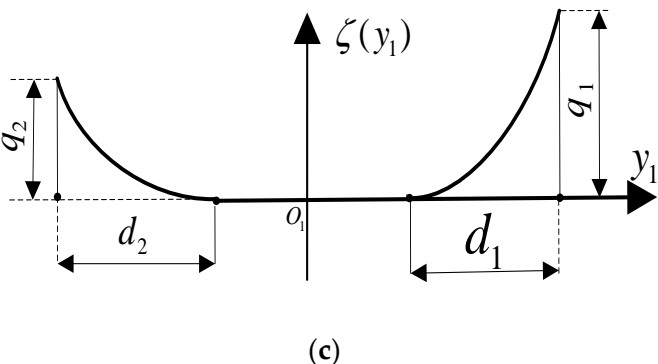

(**c**)

**Figure 2.** Ease-off design of modification surface. (**a**) The fourth-order transmission error curve. (**b**) The curved surface for the contact line modification. (**c**) The profile modification curve.

*2.3. Tooth Expression of Pinion with Ease-Off Flank Modification Both Depending on Transmission Error and Length of Contact Line*

By superimposing the normal modified surface on the tooth surface of the pinion which only contains the transmission error, the determined analytical tooth surface of pinion can be obtained. The position vector $r_{1\gamma}$ and the unit normal vector $n_{1\gamma}$ are denoted as follows:

$$
\begin{cases}
r_{1\gamma}(\mu,\beta) = \delta_1 x_1(\mu,\beta), y_1(\mu,\beta)n_1(\mu,\beta) + r_1(\mu,\beta) \\
n_{1\gamma} = \left(\frac{\partial R_1}{\partial u} + \frac{\partial \delta_1}{\partial u}n_1 + \frac{\partial n_1(\mu,\beta)}{\partial u}\delta_1\right) \times \left(\frac{\partial R_1}{\partial \beta} + \frac{\partial \delta_1}{\partial \beta}n_1 + \frac{\partial n_1(\mu,\beta)}{\partial \beta}\delta_1\right) \\
\frac{\partial n_1(\mu,\beta)}{\partial \beta} = \frac{\partial \delta_1(x_1,y_1)}{\partial x_1}\frac{\partial x_1}{\partial \beta} + \frac{\partial \delta_1(x_1,y_1)}{\partial y_1}\frac{\partial y_1}{\partial \beta}
\end{cases}
\tag{7}
$$

The ease-off flank modification of the pinion is expressed as follows:

$$
\delta(\mu,\beta) = r_{1\gamma}(\mu,\beta) - r_{10}(\mu,\beta) \cdot n_{10}(\mu,\beta)
\tag{8}
$$

## 3. Determination of Ease-Off Flank Modification Parameters

### 3.1. TCA Model for Hypoid Gears with Ease-Off Flank Modification

In [30], the TCA method of ease-off topological modification spiral bevel gear is proposed, but its model is unknown. In [31], according to the distance relationship between the mapping curve of instantaneous contact line on the ease-off topological surface and the rotating projection plane, the paper determined the contact point of the tooth surface, the contact trace, as well as transmission error. This method is basically the same as the TCA of the digital tooth surface.

In fact, the information of transmission error and contact ellipse was included in the ease-off surface. We assume that the tooth surface modification along the contact line is $\delta_{\mathrm{m}}$, min $\{\delta_{\mathrm{m}}\}$ represents the transmission error, and $\delta_{\mathrm{m}}$-min $\{\delta_{\mathrm{m}}\}$ is the normal clearance of the tooth surface, which can further verify the results of TCA. In this paper, there is a definite analytical expression for the ease-off modification tooth surface. Here, the traditional TCA method is still employed, and the expression is as follows:

$$
\begin{cases}
M_{sr}M_{rp}M_{qp}M_{pl}(\varphi_1)r_{1r}(u_1,\beta_1) - M_{s2}(\varphi_1)r_2(u_2,\beta_2) = 0 \\
r_1(u,\beta,\theta_1,) = M_{qr}M_{rs}M_{s2}(\theta_2)r_2(u_1,\beta_1) - L_{s2}(\varphi_2)L_2(u_2,\beta_2)n_2 = 0
\end{cases}
\tag{9}
$$

In the formula, $\varphi_1$ and $\varphi_2$ represent the rotation angle of active and passive gears in the meshing process of gear pair. Vector Equation (9) yield a system of five independent non-linear equations in six unknowns taking into account that $|n_{1r}| = |n_2| = 1$. Supposing $\varphi_1$ as the input one and the solution of the five non-linear equations is an iterative process. The solution of five non-linear equations discussed above is based on application of the theorem of implicit function system existence and is represented by functions

$\{u_1(\phi_1), \beta_1(\phi_1), u_2(\phi_1), \beta_2(\phi_1), \phi_2(\phi_1)\} \in C^1$. The Jacobian of the system of equations provided by vector Equation (9) has to differ from zero, and this is the precondition that surfaces are in point contact, but not in line contact [1].

### 3.2. Determination of Ease-Off Flank Modification Parameters Based on LTCA

The LTCA model employed in this study was developed by Zhang and Fang; readers are referred to [2] for further details on this model, ALTE is the direct excitation of vibration in the working process which can produced vibration and noise. Based on TCA and LTCA, the normal deformation of gear tooth in a meshing period is solved, which is transformed into the displacement on the meshing line that is the loaded transmission error. The parameters of the contact clearance between gears ($\varepsilon_1 \sim \varepsilon_4$ and $\lambda$) and the normal contact clearance between gears ($d_1, d_2, q_1, q_2$, and $\theta_a$) can be determined by optimizing the minimum loads of tooth surface and the minimum ALTE in one meshing period, respectively. The particle swarm optimization (PSO) algorithm is employed to obtain the solution. The optimization flowchart is shown in Figure 3, and the objective function is as follows:

$$f(y) = \min\{\omega G_1 / \omega G_{10} + (1 - \omega)G_2 / G_{20}\} \tag{10}$$

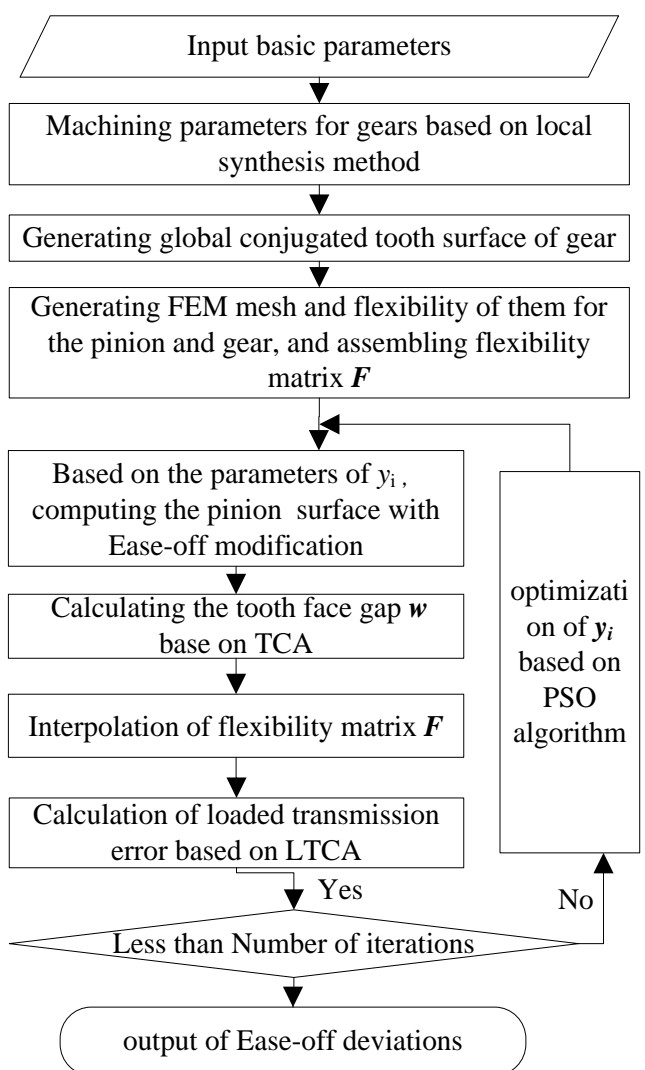

**Figure 3.** The Flowchart of the Ease-off modified optimization.

In the formula, $y$ is the optimization variable; $w$ is the weight coefficient; and $G_{10}$ and $G_{20}$ and $G_1$ and $G_2$ are the ALTE and the maximum load before and after the modification, respectively.

### 3.3. Actual Coincidence Degree of Modification Tooth Surface

ALTE is mainly determined by the effective coincidence degree (actual coincidence degree) of the teeth after being loaded. The transmission error of the loaded gear teeth is shown in Figure 4. $\Phi_p$ is the pinion angle in a meshing cycle, and $\varphi_e$ is the change in the angle from meshing in to meshing out. The coincidence degree is defined as follows:

$$\begin{cases} \varepsilon_e = \varphi_r / \varphi_p - \varepsilon_o (\varepsilon_o \geq 0) \\ \varepsilon_t = \varphi_e / \varphi_p \end{cases} \tag{11}$$

Here, the design of the coincidence degree $\varepsilon_t$ is only related to the geometric dimension of the gear, and the effective coincidence degree $\varepsilon_e$ refers to the coincidence degree of the actual contact between the teeth, which is related to the profile modification of the tooth. ① The coincidence degree reflected by the clearance between teeth can be described by the actual angle $\varphi_r / \varphi_p$. ② The coincidence degree reflected by the normal clearance along the long axis of the ellipse is represented by $\varepsilon_0$. Its value cannot be described quantitatively but only in terms of its influence on the changing trend of the actual degree of coincidence. For a tooth pair with a modified instantaneous contact line, $\varepsilon_0$ decreases gradually with the increase in loads until the gear pair makes complete contact along the contact line and then $\varepsilon_0$ is 0. When the tooth surface clearance is 0 due to the increases of load, the actual coincidence degree is equal to the theoretical coincidence degree, and reaches its maximum. That is to say $\varepsilon_e \leq \varepsilon_t$.

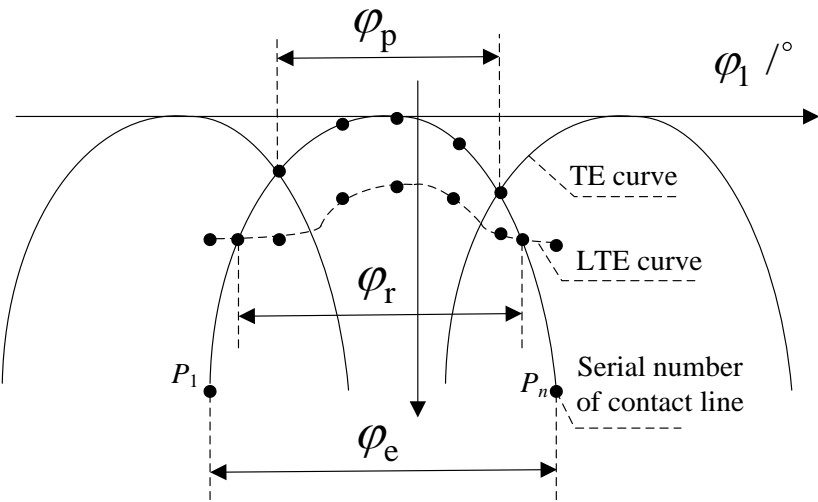

**Figure 4.** Geometrical transmission error and loaded transmission error curve.

## 4. Numerical Examples

For the high-precision gear pair, in order to increase the actual coincidence degree after loading, the design of transmission error is considered firstly to reduce the vibration, and then the topological modification tooth surface is employed to change the shape, size, and position of the contact mark. Take a pair of high precision hypoid gears as an example. See Table 1 for basic parameters and Table 2 for theoretical cutting parameters (the rated torque of gear is 600 N·m).

**Table 1.** Geometric parameters of the hypoid gear pair.

| Parameters | Pinion | Gear |
|---|---|---|
| Tooth number | 8 | 41 |
| Mean helix angle/(deg) | 48.93 | 30.63 |
| Addendum height/(mm) | 5.77 | 1.05 |
| Dedendum height/(mm) | 1.16 | 5.73 |
| Pitch cone angle/(deg) | 12.53 | 76.82 |
| Face cone angle/(deg) | 17.45 | 77.73 |
| Root cone angle(deg) | 11.67 | 71.68 |
| Outer cone distance/(mm) | 97.19 | 84.72 |
| Face width/(mm) | 28 | 24 |
| Axial setting | 13.84 | −1.43 |
| Generation type | Generated | Formate |
| Offset distance/(mm) | 23 | |
| Shaft angle/(deg) | 90 | |

**Table 2.** Machining parameters for theoretical tooth surfaces.

| Parameters | Pinion (Concave Surface) | Gear (Convex Surface) |
|---|---|---|
| Tilt angle/(deg) | 16.7 | 0 |
| Swivel angle/(deg) | 346.7 | 0 |
| Cutter radius/(deg) | 80.5 | 75.4 |
| Cutter angle/(deg) | 20 | 22.5 |
| Radial setting/(mm) | 73.95 | 74.04 |
| Basic cradle angle/(deg) | 83.6 | −66.3 |
| Vertical/(mm) | 19.6 | 0 |
| Axial/(mm) | 2.4 | 0.6 |
| Machine center to back/(mm) | 2.7 | 0 |
| Machine root angle/(deg) | −2 | 70.9 |
| Roll ratio | 5.064418 | —— |

The TCA and ease-off modified deviation are shown in Figure 5 for the theoretical tooth, the conjugate tooth, and the best ease-off tooth. The tooth contact path presents a large angle diagonal contact (see Figure 5a,b). The contacted path for the conjugate tooth surface is along the pitch circle with little transmission error (see Figure 5c).

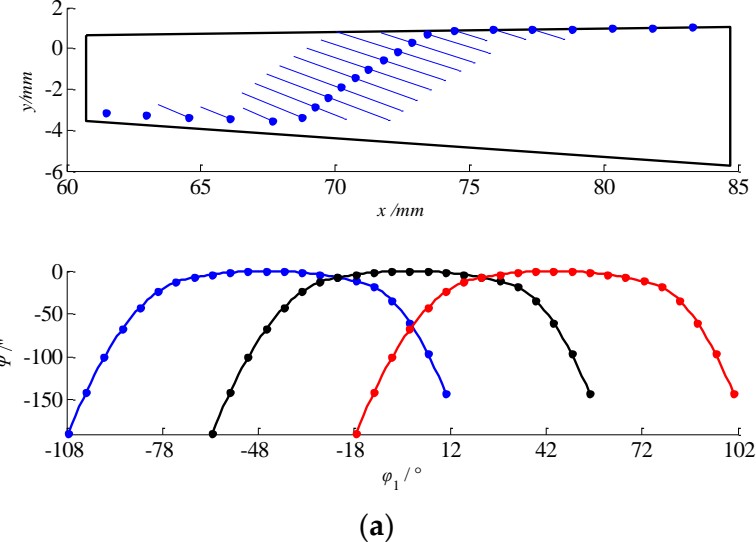

(a)

**Figure 5.** *Cont.*

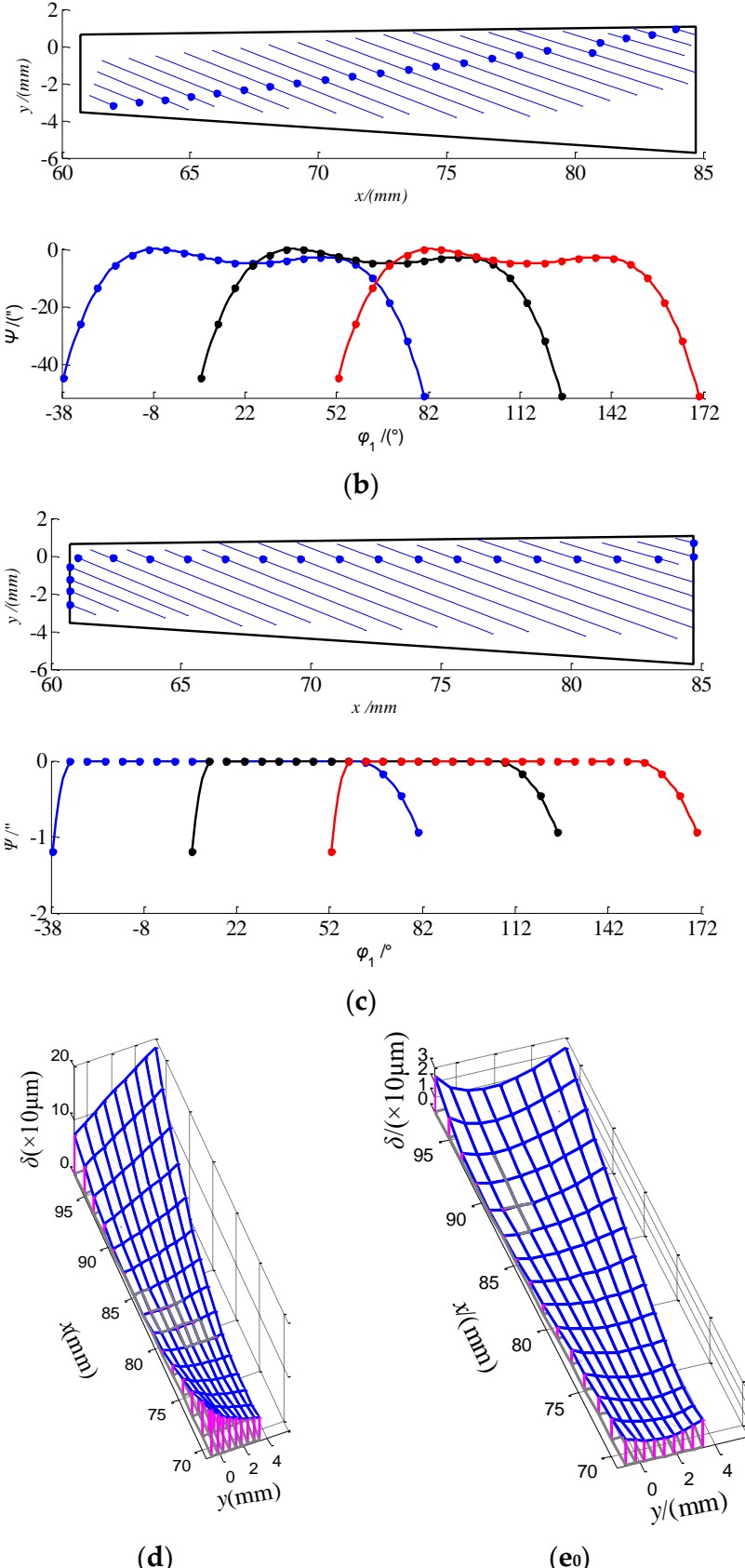

**Figure 5.** *Cont.*

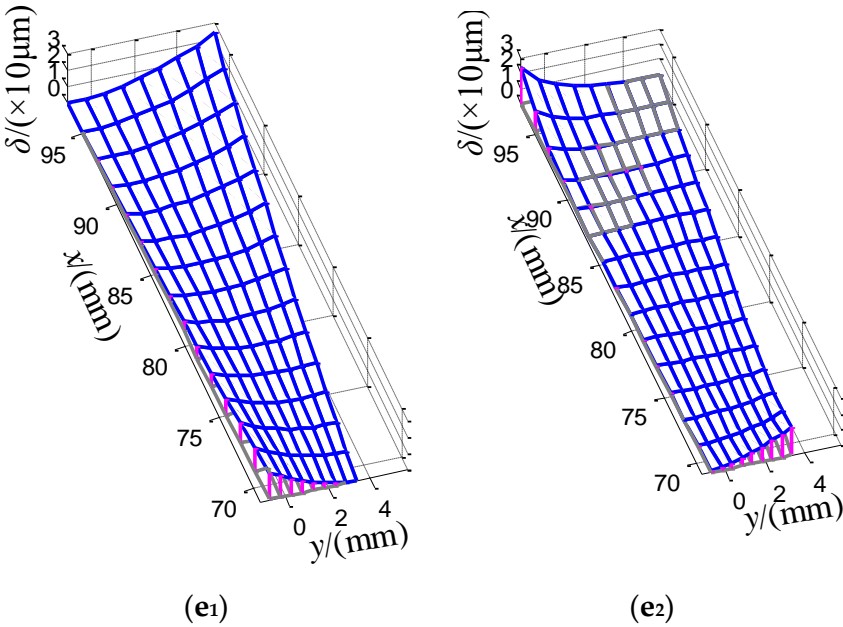

(**e₁**)　　　　　　　　　　　　　　　　　(**e₂**)

**Figure 5.** TCA simulation and ease-off deviation. (**a**) TCA for the theoretical tooth. (**b**) TCA for the best ease-off tooth. (**c**) TCA for the conjugate tooth. (**d**) The ease-off deviation for the theoretical tooth. (**e₀**) The best ease-off deviation for the optimization tooth. (**e₁**) The ease-off deviation, dependent only on the length of the instantaneous contact line. (**e₂**) The ease-off deviation, dependent only on the transmission error.

The ease-off modification of the theoretical tooth is shown in Figure 5d. The best ease-off modification of the optimized tooth is shown in Figure 5e₀, and is composed of contact gaps from the contact line of the tooth (see Figure 5e₁) and gaps of the tooth from the unload transmission error (see Figure 5e₂).

Conclusions can be drawn from Figure 5 that ease-off modification are consistent with the contact area for the theoretical tooth, the conjugate tooth, and the best ease-off tooth, respectively. The larger the amplitude of transmission error, the larger the tooth surface deviation. There are 22 contact points on the tooth surface, and a meshing period is divided into eight equal parts, and then the design of coincidence degree $\varepsilon_t$ is close to $(22 - 1)/8 = 2.6$. The ease-off modification is the largest for the theoretical tooth, which can reduce the sensitivity of assembly error, but leads to a decrease in the effective degree of contact $\varepsilon_e$. The ease-off modification refers to the zeros for the conjugate tooth, which contributes to the biggest coincidence degree, but leads to greater sensitivity to assembly error. The best ease-off modification can reduce the sensitivity of the assembly error, with an effective coincidence degree greater than the theoretical tooth.

Tooth load distribution based on LTCA is shown in Figure 6. The load distribution of the theoretical tooth is shown in Figure 6a, and is concentrated into the central. The load distribution of the conjugate tooth is shown in Figure 6b, and has obvious contact with the edge of tooth, while the load value is basically the same in the middle, top, and root of the tooth. The load distribution of the best ease-off tooth is shown in Figure 6c, and the value of loads is small in the addendum, root, heel, and toe, which could easily help to avoid stress concentration. The total loads on the contact line decreases for the best ease-off tooth and the conjugate tooth.

The load-sharing coefficients are shown in Figure 6d. The peak value of the load-sharing coefficients reaches the maximum because of excessive mismatch for the theoretical tooth. The sum of the load-sharing coefficients is equal to 1 for the contact positions 1, 9, and 17. The maximum load-sharing coefficients come to the heel, and the curve is asymmetric for the conjugate tooth, because the length of the contact line from the heel to the toe tends to decline.

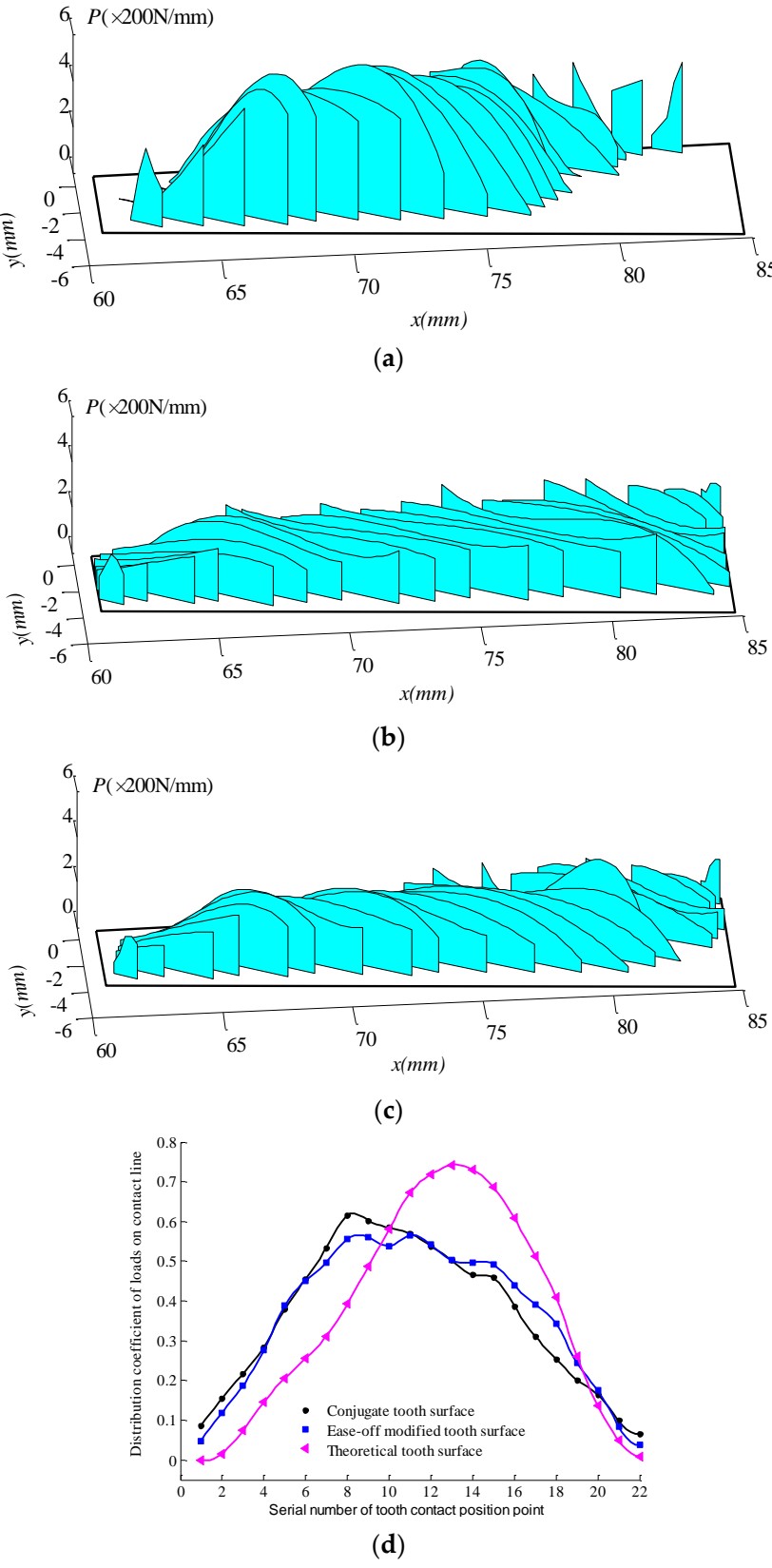

**Figure 6.** Tooth load distribution based on LTCA. (**a**) The load distribution for theoretical tooth. (**b**) The load distribution for the conjugate tooth surface. (**c**) The load distribution for the ease-off modification tooth surface. (**d**) Loads sharing a coefficient of the contact lines.

Conclusions drawn from Figure 6 show the smallest load value, which can be evenly distributed along the whole tooth for the best ease-off tooth. Besides, the best ease-off tooth avoids loads of the tooth edge.

Loaded transmission error and its amplitude under loads are shown in Figure 7. The influences of the actual coincidence degree $\varepsilon_e$ on ALTE under loads is discussed as follows:

(1) ALTE increases with gradually increasing loads for the conjugate tooth because $\varepsilon_e$ is a constant value with increasing loads and reaches the maximum value equal to $\varepsilon_t$ (Figure 7a).

(2) As the load increases, $\varepsilon_e$ approaches the maximum value gradually from a small value due to excessive tooth mismatch for the theoretical tooth. So, the curve of the ALTE value fluctuates with increasing loads for the theoretical tooth. When the load form of 300 N·m increases to 600 N·m, the contact ellipse length increases ($\varepsilon_0$ decreases). Then, $\varepsilon_e$ increases, so ALTE decreases. When the load is $\geq$600 N·m, $\varepsilon_e$ reaches the maximum, so the ALTE gradually increases (Figure 7a).

(3) The relationship between ALTE and loads for the best ease-off tooth shows that when the load increases to 300 N·m, $\varphi_r/\varphi_p$ gradually increases, so $\varepsilon_e = \varphi_r/\varphi_p - \varepsilon_0$ increases and then ALTE decreases. When the load increases from 300 to 600 N·m, the normal clearance decreases ($\varepsilon_0$ decreases). Therefore, $\varepsilon_e$ increases still and ALTE decreases. Similarly, when the load is $\geq$600 N·m, the actual coincidence $\varepsilon_e$ does not change (reaching the maximum), so the ALTE increases (Figure 7b).

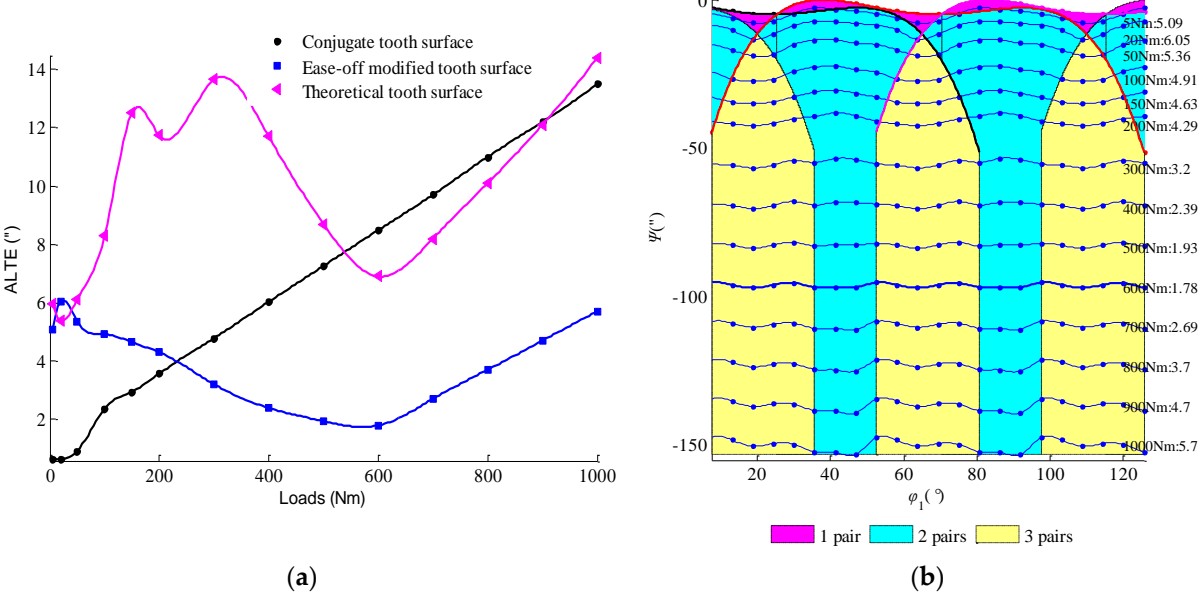

(**a**)  (**b**)

**Figure 7.** Loaded transmission error and its amplitude under loads. (**a**) Amplitude of loaded transmission error under loads. (**b**) Loaded transmission error under loads for the best ease-off tooth.

Conclusions can be drawn from Figure 7 that the actual coincidence, $\varepsilon_e$, refers to a variable value with increasing loads for the tooth with modification. ALTE decreases with increasing $\varepsilon_e$. After $\varepsilon_e$ reaches the maximum, ALTE increases with increasing loads.

## 5. Conclusions

(1) The tooth of the pinion globally conjugated with the gear is derived. The best ease-off modification parameter of the hypoid gear is determined by optimizing the minimum ALTE based on TCA and LTCA.

(2) The contact clearances and the clearances between the tooth have a great impact on the coincidence degree. Since the coincidence degree of the conjugate tooth is constant, ALTE increases with increasing loads. However, the coincidence degree of modification is variable with increasing load, so ALTE also changes curve. On the one hand, when the

actual coincidence degree increases less, the ALTE gradually increases. On the other hand, when the actual coincidence degree increases dramatically, the ALTE decreases steadily. When all the tooth clearance is completely eliminated with increasing loads, the degree of coincidence does not change any more, and the ALTE reaches the minimum. After that, as the load goes up, the ALTE increases gradually.

**Author Contributions:** J.J. proposed the key idea of this paper. Q.W. proposed the modification approach of ease-off flank for hypoid gears, based on this approach established the dynamic model and improved meshing performance of automobile drive axle. H.C. and J.G. assisted with discussing the idea and results. J.T. and Y.S. proofread the article. All authors have read and agreed to the published version of the manuscript.

**Funding:** This work was supported by the Natural Science Basic Research Plan in Shaanxi Province of China (2018JM5089) and Key Science and Technology Program of Shaanxi Province (2020NY-148; 2020GY-188) and the Project National United Engineering Laboratory for Advanced Bearing Tribology of Henan University of Science and Technology (202106); Science and Technology Project of Beilin District (GX2140); Science and Technology Project of Weiyang District (202111); Science and Technology Project of Xi'an (21XJZZ0027).

**Institutional Review Board Statement:** Not applicable.

**Informed Consent Statement:** Not applicable.

**Data Availability Statement:** Not applicable.

**Acknowledgments:** We thank sincerely the editorial board members and anonymous reviewers for their constructive comments.

**Conflicts of Interest:** The authors declare no conflict of interest.

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
