# Peer review of "Design and Analysis for Hypoid Gears with Ease-Off Flank Modification"

_applsci, doi:10.3390/app12020822_

Round 1
Reviewer 1 Report
The manuscript addresses a relevant research problem related the design of the gear tooth profile aiming to improve the transmission performance of automobile drive axles. However, the manuscript is poorly written, and the presentation of the methodology and the results sections are not well organized. Furthermore, the conclusions are loosely connected with the discussion presented in the results section. In my opinion, some of the critical points of the manuscripts that must be improved before considering it for publication are:
- The schematic illustration of the hypoid gear coordinate system presented in Fig. 1 of the manuscript is crowded of symbols, therefore, is too difficult to understand. The authors should consider including the contours of the meshing pinion/gear set in a larger scale, showing clearly where each coordinate system/dimensions/angles are located meshing pinion/gear set.
- In line 64, the authors refer to the coordinate system S2 as the “large-wheeled coordinate systems”. Please, explain the term “large-wheeled”.
- It is not clear what distances H1 and H2 represent in Figure 1. Also, in line 65-66, one can read that these parameters “are the distance from small or big round section cone vertex to the intersection respectively”. What does that mean if the figure does not show any round section cone vertex?
- The position vector r10 and the normal vector n10 of the pinion are taken with respect to which coordinate system?
- What is the function f1 shown in Eq. (2)? A clear explanation of all terms in this equation must be provided.
- The same symbol z2 is being used to denote both the number of teeth of the gear (line 62) and one of the coordinate axes of the S2 coordinate system (Fig. 1). Please, change one of them.
- The symbol j1 in the horizontal axis depicted in Fig. 2(a) should be replaced by q1.
- In the flowchart presented in Fig. 3, please provide the meaning of the PSO optimization algorithm. Also, the “Yes” should be moved to below the bifurcation diamond.
- The presentation and discussion of the results are the main flaw of the paper, in my opinion. All results from Fig. 5 to Fig. 8 must be introduced first, in a brief explanation addressing the “what?” question (in other words, what result this figure is presenting?), followed by the answer to the “why?” question (why is the result presented by the figure is relevant?), and ending by the answer to the “so what?” question (i.e., what conclusions can be drawn from that figure in the context of the hypothesis tested by the paper).
- The captions and the results shown in Figs. 7 and 8 are not consistent with the discussion presented in lines 216-232.
- The conclusion of the paper needs to be rewritten. Some of the listed items were taken from the methodology section.
- How the proposed model can be validated? Although a full validation of the proposed method might be out of the scope of this manuscript, the authors should provide insights towards ideas to validate their proposed method.
- About half of the cited references were published about 10 years ago or more. The authors should consider updating their referenced by newer ones showing the state-of-the-art of the current gear design methods.
Author Response
Comments and Suggestions for Authors
The manuscript addresses a relevant research problem related the design of the gear tooth profile aiming to improve the transmission performance of automobile drive axles. However, the manuscript is poorly written, and the presentation of the methodology and the results sections are not well organized. Furthermore, the conclusions are loosely connected with the discussion presented in the results section. In my opinion, some of the critical points of the manuscripts that must be improved before considering it for publication are:
- The schematic illustration of the hypoid gear coordinate system presented in Fig. 1 of the manuscript is crowded of symbols, therefore, is too difficult to understand. The authors should consider including the contours of the meshing pinion/gear set in a larger scale, showing clearly where each coordinate system/dimensions/angles are located meshing pinion/gear set.
Answer:thank you for your suggestion. I have revised the manuscript,And the revised sentences are shown in blue font in the manuscript.
Coordinate system of gear and pinion meshing in Fig.1 in detailed are shown in Refs.17
- In line 64, the authors refer to the coordinate system S2 as the “large-wheeled coordinate systems”. Please, explain the term “large-wheeled”.
Answer:thank you for your suggestion. the “large-wheeled coordinate systems” correct to the “gear coordinate systems”
- It is not clear what distances H1and H2represent in Figure 1. Also, in line 65-66, one can read that these parameters “are the distance from small or big round section cone vertex to the intersection respectively”. What does that mean if the figure does not show any round section cone vertex?
Answer:thank you for your suggestion. Delete “are the distance from small or big round section cone vertex to the intersection respectively” and add “H1, H2 are the axial setting”.
- The position vector r10 and the normal vector n10of the pinion are taken with respect to which coordinate system?
Answer:thank you for your suggestion. I have revised the manuscript,And the revised sentences are shown in blue font in the manuscript.
r10 and n10 are the position vector and normal vectors of pinion globally conjugated with the gear, and r10 are taken with respect to S1.
- What is the function f1shown in Eq. (2)? A clear explanation of all terms in this equation must be provided.
Answer:thank you for your suggestion. I have revised the manuscript,And the revised sentences are shown in blue font in the manuscript.
f1 is called the equation of meshing. f1=n10•v10,where v10 is the relative velocity between the cutter blade (the pinion)and the work gear in coordinate system S1.
- The same symbol z2 is being used to denote both the number of teeth of the gear (line 62) and one of the coordinate axes of the S2coordinate system (Fig. 1). Please, change one of them.
Answer:thank you for your suggestion. I have revised the manuscript,And the revised sentences are shown in blue font in the manuscript.
- The symbol j1in the horizontal axis depicted in Fig. 2(a) should be replaced by q1.
Answer:thank you for your suggestion. I don't understand your suggestion
- In the flowchart presented in Fig. 3, please provide the meaning of the PSO optimization algorithm. Also, the “Yes” should be moved to below the bifurcation diamond.
Answer:thank you for your suggestion. I have revised the manuscript,And the revised sentences are shown in blue font in the manuscript.
- The presentation and discussion of the results are the main flaw of the paper, in my opinion. All results from Fig. 5 to Fig. 8 must be introduced first, in a brief explanation addressing the “what?” question (in other words, what result this figure is presenting?), followed by the answer to the “why?” question (why is the result presented by the figure is relevant?), and ending by the answer to the “so what?” question (i.e., what conclusions can be drawn from that figure in the context of the hypothesis tested by the paper).
Answer:thank you for your suggestion. I have revised the manuscript,And the revised sentences are shown in blue font in the manuscript.
- The captions and the results shown in Figs. 7 and 8 are not consistent with the discussion presented in lines 216-232.
Answer:thank you for your suggestion. I have revised the manuscript,And the revised sentences are shown in blue font in the manuscript.
- The conclusion of the paper needs to be rewritten. Some of the listed items were taken from the methodology section.
Answer:thank you for your suggestion. I have revised the manuscript,And the revised sentences are shown in blue font in the manuscript.
- How the proposed model can be validated? Although a full validation of the proposed method might be out of the scope of this manuscript, the authors should provide insights towards ideas to validate their proposed method.
Answer:thank you for your suggestion.There is no experimental validation due to lack of funding.
- About half of the cited references were published about 10 years ago or more. The authors should consider updating their referenced by newer ones showing the state-of-the-art of the current gear design methods.
Answer:thank you for your suggestion, and some new literature are supplemented
Reviewer 2 Report
An interesting paper has been submitted for consideration. The proposed method of ease-off modification is interesting and rather well described. I have only three remarks, listed below:
1) English language is rough in some places. Sometimes the wrong tense is used. Generally, the paper should be checked by a native speaker. Please avoid starting the sentence from the symbol as it is done several times in the paper.
2) Please give the definition of each matrix from eq (2). Moreover, please provide the details about the solution of (9).
3) In conclusions Authors should compare their method with existing ones in terms of contact pattern and amplitude (for example with gear pair designed with commercial software from Gleason or Klingelnberg).
The above comments yield the review to recommend the paper for publication after minor revisions.
Author Response
An interesting paper has been submitted for consideration. The proposed method of ease-off modification is interesting and rather well described. I have only three remarks, listed below:
1) English language is rough in some places. Sometimes the wrong tense is used. Generally, the paper should be checked by a native speaker. Please avoid starting the sentence from the symbol as it is done several times in the paper.
Answer:thank you for your suggestion. I have revised the manuscript, and the revised sentences are shown in blue font in manuscript.
- Please give the definition of each matrix from eq (2). Moreover, please provide the details about the solution of (9).
Answer:thank you for your suggestion. I have revised the manuscript, and the revised sentences are shown in blue font in manuscript.
- In conclusions Authors should compare their method with existing ones in terms of contact pattern and amplitude (for example with gear pair designed with commercial software from Gleason or Klingelnberg).
The above comments yield the review to recommend the paper for publication after minor revisions.
Answer:Thank you for your suggestion. There are not edge contact of tooth in terms of contact pattern for commercial software of Gleason or Klingelnberg).
Reviewer 3 Report
Article deals with design and analysis for hypoid gears. The paper is processed clearly and at high scientific soundness.
The study is very interesting and i thing that paper may be accepted after some minor revisions.
- I am asking the authors to specify in more detail, within the Abstract, the results obtained during the research, which is presented in the article.
- Although the Introduction is written well, I propose to be developed in more detail; especially with regard to the cited individual publications:"4-6, 11-13, 14-16"
Author Response
Comments and Suggestions for Authors
Article deals with design and analysis for hypoid gears. The paper is processed clearly and at high scientific soundness. The study is very interesting and i thing that paper may be accepted after some minor revisions.
- I am asking the authors to specify in more detail, within the Abstract, the results obtained during the research, which is presented in the article.
Answer:thank you for your suggestion. I have revised the manuscript, and the revised sentences are shown in blue font in manuscript.
- Although the Introduction is written well, I propose to be developed in more detail; especially with regard to the cited individual publications:"4-6, 11-13, 14-16"
Answer:thank you for your suggestion. I have revised the manuscript, and the revised sentences are shown in blue font in manuscript.
Round 2
Reviewer 1 Report
The reviewer wants to thank the authors for implementing all suggestions. however, the manuscript still needs to go through a thorough English grammar and formating review before publication.